# Heterogeneity and transcriptome changes of human CD8+ T cells across nine decades of life

Jian Lu[1,7], Raheel Ahmad[1,7], Thomas Nguyen [1], Jeffrey Cifello[1], Humza Hemani[1], Jiangyuan Li [1], Jinguo Chen[2], Siyi Li[1], Jing Wang[1], Achouak Achour[1], Joseph Chen[1], Meagan Colie [1], Ana Lustig[1], Christopher Dunn [1], Linda Zukley[3], Chee W. Chia[4], Irina Burd [5], Jun Zhu [6], Luigi Ferrucci [3] & Nan-ping Weng [1] ✉

The decline of CD8+ T cell functions contributes to deteriorating health with aging, but the mechanisms that underlie this phenomenon are not well understood. We use single-cell RNA sequencing with both cross-sectional and longitudinal samples to assess how human CD8+ T cell heterogeneity and transcriptomes change over nine decades of life. Eleven subpopulations of CD8+ T cells and their dynamic changes with age are identified. Age-related changes in gene expression result from changes in the percentage of cells expressing a given transcript, quantitative changes in the transcript level, or a combination of these two. We develop a machine learning model capable of predicting the age of individual cells based on their transcriptomic features, which are closely associated with their differentiation and mutation burden. Finally, we validate this model in two separate contexts of CD8+ T cell aging: HIV infection and CAR T cell expansion in vivo.

CD8+ T cells are a heterogeneous population that collectively fulfills the antigen-specific cytotoxic function in the immune response. After first encountering an antigen presented by dendritic cells, antigen-specific naïve CD8+ T cells proliferate and differentiate to become effector cells and memory cells[1,2]. Differentiation is associated with epigenetic and transcriptomic changes leading to the gain of distinct functions such as rapid cytokine production, cytotoxicity by effector cells during recall response, and longevity in memory cells[3–5]. The characterization of human CD8+ T cells by flow cytometry has identified naïve, effector, and various types of memory cells, providing the framework to understand the composition and function of CD8+ T cells[6–8]. However, the extent of CD8+ T cell heterogeneity and its

implication in the aging of immune function has not been fully examined.

Aging alters the composition and functions of CD8+ T cells[9,10]. In older adults, reduced numbers of circulating naïve CD8+ T cells and increased numbers of terminally differentiated memory CD8+ T cells are frequently observed[11–13]. As a result, the functional decline of the CD8+ T cell response to infectious agents or cancerous cells has been reported[14]. This decline is attributed to two broad age-related changes: extrinsic changes related to the microenvironment where CD8+ T cells are maintained and the immune response is initiated, and intrinsic changes related to the capability of CD8+ T cells to respond to stimulation-induced proliferation, differentiation, and delivery of

[1]Laboratory of Molecular Biology and Immunology, National Institute on Aging, National Institutes of Health, Baltimore, MD, USA. [2]Center for Human Immunology, Autoimmunity and Inflammation, National Institutes of Health, Bethesda, MD, USA. [3]Translational Gerontology Branch, National Institute on Aging, National Institutes of Health, Baltimore, MD, USA. [4]Laboratory of Clinical Investigation, National Institute on Aging, National Institutes of Health, Baltimore, MD, USA. [5]Integrated Research Center for Fetal Medicine, Department of Obstetrics and Gynecology, Johns Hopkins University School of Medicine, Baltimore, MD, USA. [6]National Heart, Lung, and Blood Institute, National Institutes of Health, Bethesda, MD, USA. [7]These authors contributed equally: Jian Lu, Raheel Ahmad. ✉e-mail: Wengn@mail.nih.gov

effector molecules[15,16]. Intrinsic changes characterized by comparative transcriptomic analyses of CD8+ T cells from young and old study participants using microarray and RNA sequencing (RNAseq) methods have shown altered gene expression related to T cell receptor signaling, the cytokine/chemokine network, cytotoxicity, and metabolic preference[17–21]. These age-related transcriptomic changes reflect average differences between younger and older study participants, and therefore lack details about whether age-related changes are due to a change in the percentage of cells expressing a given gene, a change in the expression level in individual cells or both. The recent development of the single-cell RNA sequencing (scRNAseq) method enables the characterization of transcriptomic profiles at the individual cell level and can indicate potential heterogeneity overlooked by bulk cell approaches[22]. Using scRNAseq, Mogilenko et al. showed increased expression of two granzymes (GZMK and GZMB) in CD8+ T cells from older compared to younger individuals[23] However, the rates and mechanisms behind such age-related gene expression changes remain unknown.

Determining a person's biological age has research and clinical application[24] and is currently assessed at the cell level by measures that include telomere length[25], alterations of DNA methylation patterns[26,27] and number of somatic mutations[28,29]. Recently, age-related transcriptomic changes in blood cells have been explored to estimate cell age and have showed a close correlation with chronological age[30,31]. These transcriptome-based age predictions rely on predominantly altered genes with age, but recent applications of machine learning (ML) algorithms have integrated more subtly changed genes to successfully predict the age of individuals from the transcriptomic profiles of dermal fibroblasts[32]. Biological age estimates based on the transcriptomic features of study participants are often informed by sampling at a single time point, which results in a high degree of individual variation. Longitudinal analysis of age-related changes by sampling the same measures over time from the same participants overcomes this weakness and offers a more accurate measure of biological aging. Using flow cytometry and RNAseq, Alpert et al. analyzed a longitudinal cohort and showed a high-dimensional trajectory of immune system age[33]. However, heterogeneity in the biological age of individual immune cells from individuals of different ages has not been examined.

In this study, we analyze individual CD8+ T cells from three cohorts of healthy human participants using high dimensional flow cytometry (4,600,000 cells from 165 donors) and scRNAseq (120,418 cells from two cohorts of healthy participants: 16 who made a single blood donation and 8 who made 2-3 donations over an average of 9 years). We identify 11 subpopulations of CD8+ T cells and their shared age-related changes in gene expression. These age-related gene expression changes display three distinct modes: a change in the number of expressing cells, a change in expression level within single cells, or both. We also develop a mixed-effect elastic net (MEEN) ML algorithm to predict the age of single CD8+ T cells based on their transcriptomes with strong correlation with chronological age in cross-validation. Predicted cell ages indicate that naïve cells were younger than memory cells, positively correlated with the number of somatic mutations in single cells and indicate the changes in CD8+ T cell aging that occur in HIV infection and CAR T cell expansion in vivo. Overall, our findings identify subpopulations of CD8+ T cells and their precise changes with age in both composition and transcriptional profiles. Our results show highly predictive transcriptional features of single cells that can be used to accurately estimate cell age.

## Results

### Identification of CD8+ T cell subpopulations by scRNAseq and high-dimensional flow cytometry

To investigate the heterogeneity of circulating CD8+ T cells in humans, we used two single-cell approaches – scRNAseq and high-dimensional flow cytometry – to analyze three human cohorts (Fig. 1a). Using

scRNAseq, we analyzed cells from 24 donors in two cohorts: the first consisted of 16 individuals (8 males and 8 females) ranging from newborn to 90 years. The second cohort was 8 individuals (4 males and 4 females) ranging from age 30 to 69 years at first donation with an average of 9 years from first to last donation (Supplementary Fig. 1a). CD8+ T cells were isolated from peripheral blood mononuclear cells (PBMC) with approximately 3000-6000 cells used for droplet-based scRNAseq that constructed and sequenced single-cell gene expression libraries to greater than 80% saturation, for a total of 120,418 individual CD8+ T cells analyzed after filtering (Fig. 1a and Supplementary Fig. 1b). By flow cytometry, we analyzed cells from 165 donors (83 males, 82 females), from newborn to 96 years (Supplementary Fig. 1a). PBMCs were stained with a panel of 15 antibodies and a viability dye. CD3+CD8+ T cells were manually gated and a total of 4.6 million CD8+ T cells were analyzed after filtering.

Using scRNAseq data, 120,418 CD8+ T cells were divided into 10 clusters that were further annotated by expression of known marker genes into 6 known CD8+ T cell subsets, primarily naïve cells, including two subpopulations: adult naive ($T_{Na}$) and cord blood naïve ($T_{Nc}$), plus four memory subsets including memory stem cell ($T_{SCM}$), central memory ($T_{CM}$), three subpopulations of effector memory ($T_{EM1-3}$), two subpopulations of terminally differentiated effector memory cells ($T_{EMRA1-2}$), and an effector cell subset ($T_{EFF}$) (Fig. 1b and Supplementary Table 1). $T_{Na}$ and $T_{Nc}$ cells shared common naïve T cell markers such as CCR7, but $T_{Nc}$ cells additionally expressed fetal development-related genes such as HBA2 (Fig. 1c Supplementary Data 1). Three $T_{EM}$ subpopulations were identified by the shared expression of cytotoxic factors (GZMA, CCL5 and KLRB1) but each subpopulation preferentially expressed specific genes (Fig. 1c, Supplementary Fig. 1d, Supplementary Data 1). Finally, two $T_{EMRA}$ subpopulations (confirmed by CD45RA surface expression by a barcode antibody) expressed GZMB and KLRG1 (Fig. 1c and Supplementary Fig. 1c). All clusters contained cells from multiple donors (Supplementary Fig. 2a, b), eliminating the possibility that the identified cell types and expression patterns resulted from donor-specific or batch effects. Collectively, this scRNAseq analysis implicated heterogeneity in naïve and memory CD8+ T cells, including their distinct gene expression and potential functions (Supplementary Fig. 3, Supplementary Data 2, Supplementary Table 2).

High-dimensional flow cytometry analysis of 4.6 million CD8+ T cells from 165 donors showed 10 clusters based on 15 antibody staining profiles using the FlowSOM algorithm (Fig. 1d and Supplementary Fig. 1e)[34]. Clusters were assigned as two naïve subpopulations: $T_{Na}$ and $T_{No}$ (for older), $T_{SCM}$, $T_{CM}$, $T_{EM1-3}$, $T_{EMRA1-2}$, and $T_{EFF}$ (Fig. 1e and Supplementary Fig. 1e). We computed Spearman's rank correlations between the mean fluorescent intensity of 15 protein markers from flow cytometry and the average expression levels of the corresponding genes from scRNAseq for each subpopulation. We found that good correlations between subpopulations identified by the two methods in six subpopulations (Fig. 1f). Two subpopulations (Nc and No) only found by scRNAseq and Flow cytometry method, respectively; and three subpopulations (Em1, EM3, and EFF) did not have good correlations between these two methods.

### Common age-associated transcriptome changes in CD8+ T cell subpopulations

To investigate how CD8+ T cell subpopulations changed with age, we analyzed correlations between age and the proportions of subpopulations and their gene expression. We calculated the percentage of cells belonging to each subpopulation identified by the two single-cell methods for each donor and used linear regression to identify subpopulation changes with aging. The overall age-associated changes in CD8+ T cell subpopulations were quite comparable between the two methods (Fig. 2a, Supplementary Table 1, Supplementary Data 2a). $T_{Na}$ cells significantly decreased in percentage of CD8+ T cells with age, whereas several memory subpopulations ($T_{SCM}$, $T_{EM3}$, $T_{EMRA1}$ and

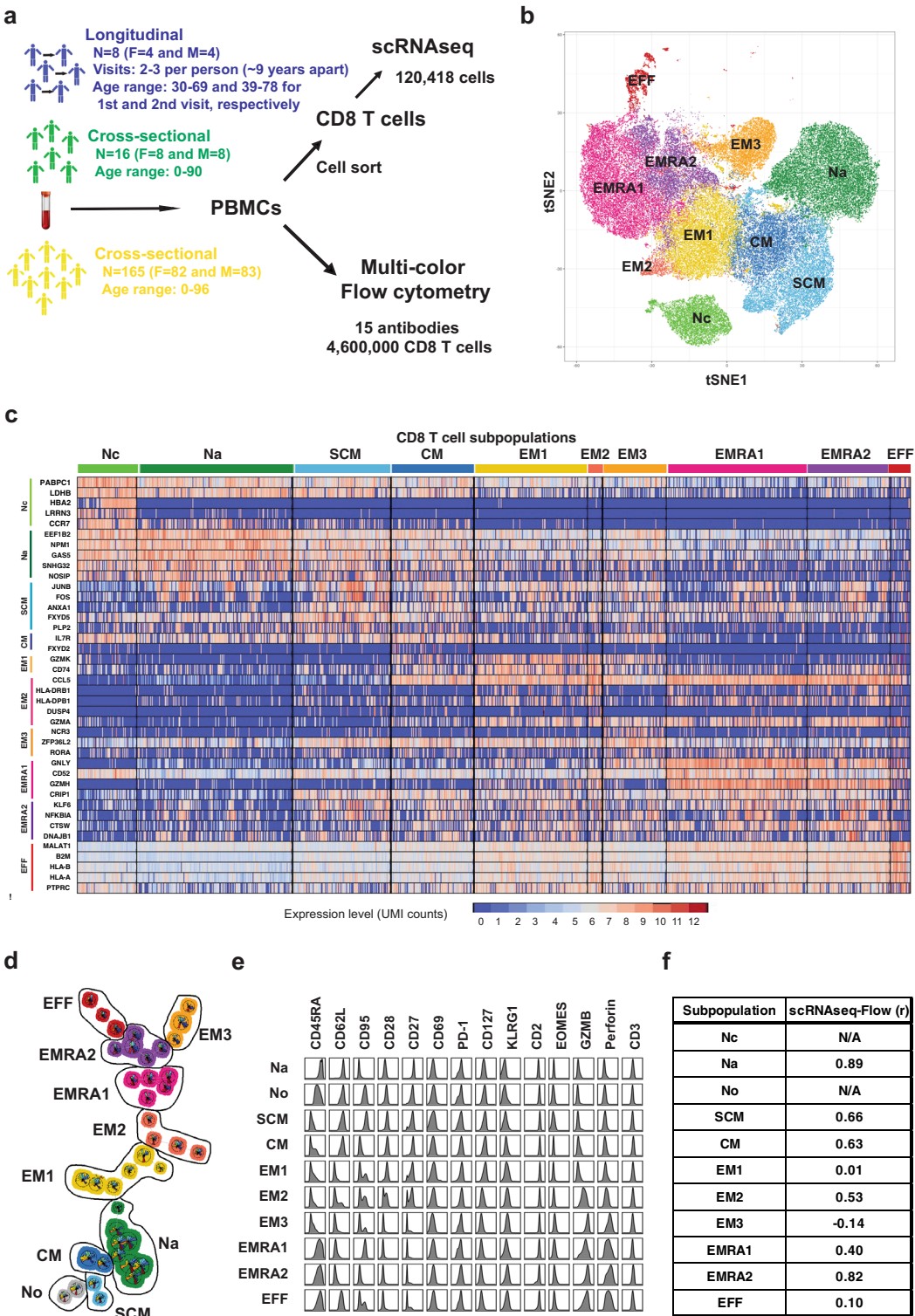

$T_{EMRA2}$) increased in percentage with age. The percentages of $T_{CM}$, $T_{EM}$, and $T_{EFF}$ remained stable over time. The age-related changes were all significant in all subpopulations except $T_{CM}$ in the flow cytometry analysis, whereas only changes in $T_{Na}$ and $T_{SCM}$ were significant in scRNAseq analysis. This difference in statistical significance was likely due to the difference in cohort size (flow cytometry cohort had 165 participants) while the scRNAseq cohort had only 24 participants). The consistency between the percentage change with age in subpopulations identified by both approaches further supported their utility as analogous techniques to study heterogeneity within CD8$^+$ T cells.

Next, we investigated how the transcriptome changed with age in the CD8$^+$ T cell subpopulations. We used MELR analysis (see Eq. 1 in method) to identify genes whose expressions changed significantly with age in each subpopulation (Fig. 2b, Supplementary Fig. 4, Supplementary Data 2b). In most subpopulations, more genes increased expression with age (1590 unique genes across nine subpopulations) than decreased (953 unique genes across nine subpopulations). We defined a significantly age-changed gene using beta coefficient (CO) > 0.0019, representing an expression change of 10% across the span of the donors' age range in either direction,

**Fig. 1 | Identification of CD8⁺ T cell subpopulations in humans by scRNAseq and high-dimensional flow cytometry. a** Experimental design. Fasting blood samples were from 24 individuals in two cohorts: 16 (aged newborn to 90) for cross-sectional and 8 (aged 30 to 69 at first donation, average 9 years to second donation) for longitudinal analysis for single-cell RNAseq (scRNAseq); and 165 (aged newborn to 96) for multicolor flow cytometry. CD8⁺ T cells were used for gene expression analysis. Peripheral blood mononuclear cells were used for flow cytometry with CD8⁺ T cells gated by antibodies and dyes. **b** t-SNE projection of 120,418 CD8⁺ T cells clustered by gene expression patterns. CD8⁺ T cell subpopulations were by comparing differentially expressed genes to previous analyses[54] and our unpublished microarray data. **c** Heatmap depicting differentially expressed genes by cluster (selected by log2FC > 0.1, FDR < 0.05; full gene list, Supplementary Data 1). **d** FlowSOM map of CD8⁺ T cell subpopulations from flow cytometry. Node sizes represent relative number of cells with similar expression patterns. Nodes are arranged into populations[34]; proximity indicates similar expression. Number of nodes is proportional to number of the population's cells. Clusters were assigned using canonical markers. **e** Expression histograms for 14 markers per CD8⁺ T cell cluster from flow cytometry. **f** CD8⁺ T cell subpopulations identified by gene expression with matched counterparts from flow cytometry identified by Spearman's rank correlation of markers and associated genes between flow cytometry and scRNA-seq clusters. Clusters within major subsets (N, CM, EM, RA) were further classified by correlation of change in percentage with age in each donor (Fig. 2a, b). F female, M male, UMI unique molecular identifier, Nc naïve cord blood, Na naive adult, SCM stem cell memory, CM central memory, EM1 2, 3, effector memory subpopulations, EMRA1 2 terminally differentiated effector memory cell sub-population, EFF effector cells, t-SNE t-distributed stochastic neighbor-embedding.

and false discovery rate <0.05. Gene set enrichment analysis (GSEA) for age-related gene expression changes identified three core enriched functional sets shared by the four major subsets of CD8⁺ T cells ($T_{Na}$, $T_{SCM/CM}$, $T_{EM1-3}$, and $T_{EMRA1-2}$): 1) antigen presentation and cellular cytotoxicity, 2) activation and exocytosis, and 3) transcription and translation (Fig. 2c). Our findings showed genes important for one functional group overlapped with the other two groups. Collectively, these findings implicated common age-associated transcriptome changes in CD8⁺ T cells and their potential functional consequences in aging cells.

## Three distinct modes of age-associated gene expression changes and related functions in CD8⁺ T cell subpopulations

After identifying average age-associated changes in gene expression in nine CD8⁺ T cell subpopulations, we examined how these changes were mechanistically achieved at the individual cell level as recently described[35]. We investigated if age-associated gene expression changes at the population level could be classified further into three distinct modes: 1) a change in the percentage of cells that express a gene in a population without alteration of its expression level in individual cells; 2) a change in the expression level of a gene in cells without a change in the total percentage of expressing cells; and 3) a combination of changes in both percentage and expression level. We analyzed genes that significantly changed across aging and observed all three modes (Fig. 3a and Supplementary Fig. 5a). Next, we grouped genes by whether they changed by a magnitude of at least 5% in either percentage or expression level. Using these criteria, of genes whose expression increased with age in nine subpopulations, 46% increase in percentage, 12% in expression level, and 14% in both (Fig. 3b, Supplementary Fig. 5b, Supplementary Data 3), with 28% of genes not meeting either classification.

To investigate if mode changes at the mRNA level also translated to the protein level, we analyzed protein level changes in granzyme A and β2-microglobulin by flow cytometry and found the parallel changes in proteins (Fig. 3c). To understand the functional changes associated with these distinct age-related modes of change, we performed GSEA for age-related gene expression changes in four major subsets of CD8⁺ T cells. We found genes that increased in percentage were related to exocytosis, oxidant detoxification, and oxidative phosphorylation. Genes that increased in expression level were associated with antigen processing and presentation as well as mRNA catabolic processes (Fig. 3d, Supplementary Data 3d).

The three distinct modes of gene expression changes were also observed in age-related reduced gene expressions (Fig. 4a). However, of all genes whose expression decreased with age in nine subpopulations: only 3% decreased in percentage, 24% in expression level, 20% in both, and 42% did not meet either classification (Supplementary Data 4). Functionally, genes that decreased in aging by mode of expression change were associated with reduced DNA replication and chromatin remodeling (Fig. 4c). Together, these findings implied three distinct modes of gene expression changes with age observed in

individual CD8⁺ T cells and diverging functional consequences associated with each mode of change.

## Prediction of cell age based on the transcriptome by a mixed effect ML algorithm

Bulk cell transcriptome-based age prediction has been recently reported[32]. Therefore, we investigated if ML models could use single-cell transcriptome data to report a biological age for a single cell that correlated with the chronological age of its donor. We applied mixed-effect modeling to integrate both longitudinal changes and cross-sectional differences into a unified aging model. To estimate the age of single CD8⁺ T cells we tested two types of ML models, mixed effects elastic net (MEEN) as a linear model and mixed-effects random forest (MERF) as a nonlinear model. We found that ages of individual CD8⁺ T cells displayed a distribution around the chronological age of the donors by MEEN (Fig. 5a, Supplementary Fig. 5a, b, Supplementary Data 5a) and MERF models (Supplementary Fig. 5c, e). Furthermore, both models predicted cell age with a significant average difference between cells from older and younger donors using both a cross-validation prediction dataset and an independent dataset (see Methods for details) (Fig. 5b MEEN, Supplementary Fig. 5d MERF). Among the two models, MEEN model had a lower root mean square error than the MERF model with the independent dataset, which represented higher accuracy on unseen data. The relative contributions of genes in each ML model were identified using a permutation-based method (methods) (Supplementary Data 5b). We then compared the top 300 age-associated genes identified by MELR across all CD8 cells with those identified by MEEN or by MERF models. Noticeably, the overlaps were relatively low at 21% (10% of increased plus 11% of decreased genes of top 300 genes) between MELR and MEEN (Fig. 5c, Supplementary Data 5c) and 4% between MELR and MERF (Supplemental Fig. 7, Supplementary Data 5c). Further analysis showed that unique genes identified by the MEEN model had smaller changes in magnitude of expression level (average CO = +/−0.001) than genes only identified by MELR model (average CO = +/−0.011) which required a minimum change of expression level (CO) > 0.0019) for us to call a gene significant (Fig. 5d).

## Age-related transcriptome changes associated with differentiation and cell divisions

To explain the underlying mechanism by which the transcriptome-based cell age was distributed around the chronological age of the cell donors, we hypothesized that the range of estimated cell ages reflected the heterogeneous nature of their true biological age. Since this heterogeneity is rooted in the proliferation of T cells, we hypothesized it would be reflected in differentiation state and mutation burden. Differentiation of CD8⁺ T cells is associated with cell division, and we found that naïve T cells were consistently predicted to be younger than memory cells for all donors (Fig. 6a, Supplementary Data 6a, b). Next, we developed a unique molecular identifier (UMI) correction-based single-cell variant calling pipeline using the genome analysis toolkit

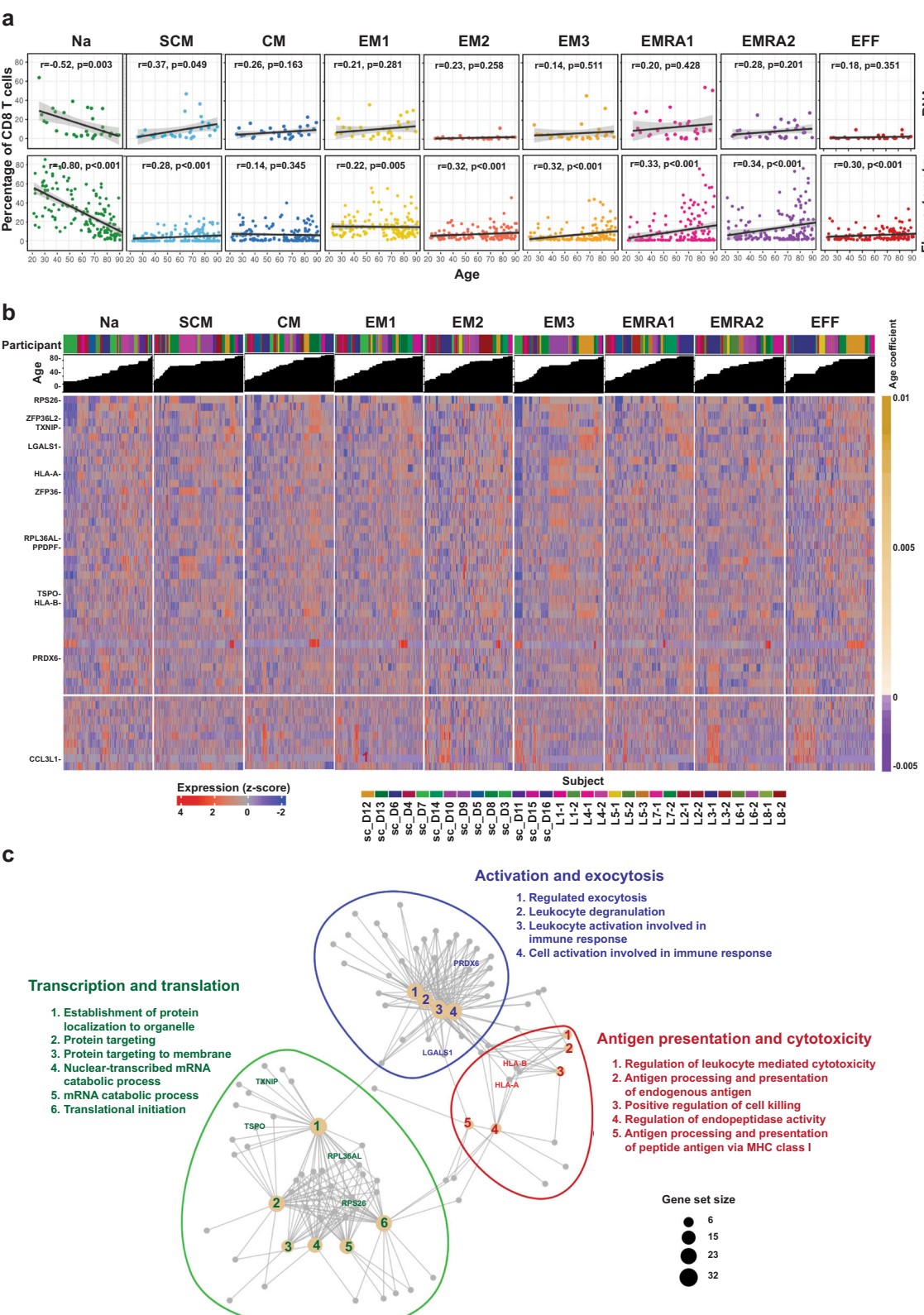

(GATK) mutect2 program[36] to investigate if mutation burden was increased in cells with older predicted age. We used scRNAseq data based on an individual's own exome sequences to call mutations, and mutations were further adjusted and rescaled to remove differences in coverage and UMI count in both single cells and each donor's exome sequencing (Supplementary Fig. 6a–e). We compared predicted cell ages to the mutation burden in single cells and found significant

positive correlations between mutation burden and predicted cell age in all four major CD8+ T cell subsets (Fig. 6b, Supplementary Data 6a, b).

## Aging of CD8+ T cells under clinical conditions quantified by the ML algorithm

Next, we examined if our MEEN model could quantify known biological changes in CD8+ T cells associated with HIV infection and CAR-T cell

**Fig. 2 | Age changes in distribution and gene expression in CD8⁺ T cell sub-populations. a** Percentages of 9 subpopulations in 24 donors as a function of age based on scRNA-seq (top panel). Trend lines were from a mixed-effect linear regression (MELR) model. Percentages of clusters for 165 donors as function of age from flow cytometry (bottom panel). Trend lines were from the MELR model for which the t distribution with n-2 degrees of freedom was used to evaluate if a slope was significant for a particular variable (2 sided t-test). **b** Expression heatmap of the top 50 most-changed genes with age shared by at least 5 of 9 subpopulations of CD8⁺ T cells based on scRNA-seq. Top panel, age-increased genes (35 genes);

bottom panel, age-decreased genes (15 genes). Full gene list is in Supplementary Data 2b and each data point was the average of 10 cells. **c**. Network plot of functional changes associated with age and associated genes (Yellow dot, functional gene set; gray dot, core enriched gene in gene set). Genes belonging to a functional set are connected by edges to that function. Nc naïve cord blood, Na naive adult, SCM stem cell memory, CM central memory, EM1, 2, 3, effector memory sub-populations 1, 2, 3, EMRA1 2 terminally differentiated effector memory cell sub-population 1, 2, EFF effector cells.

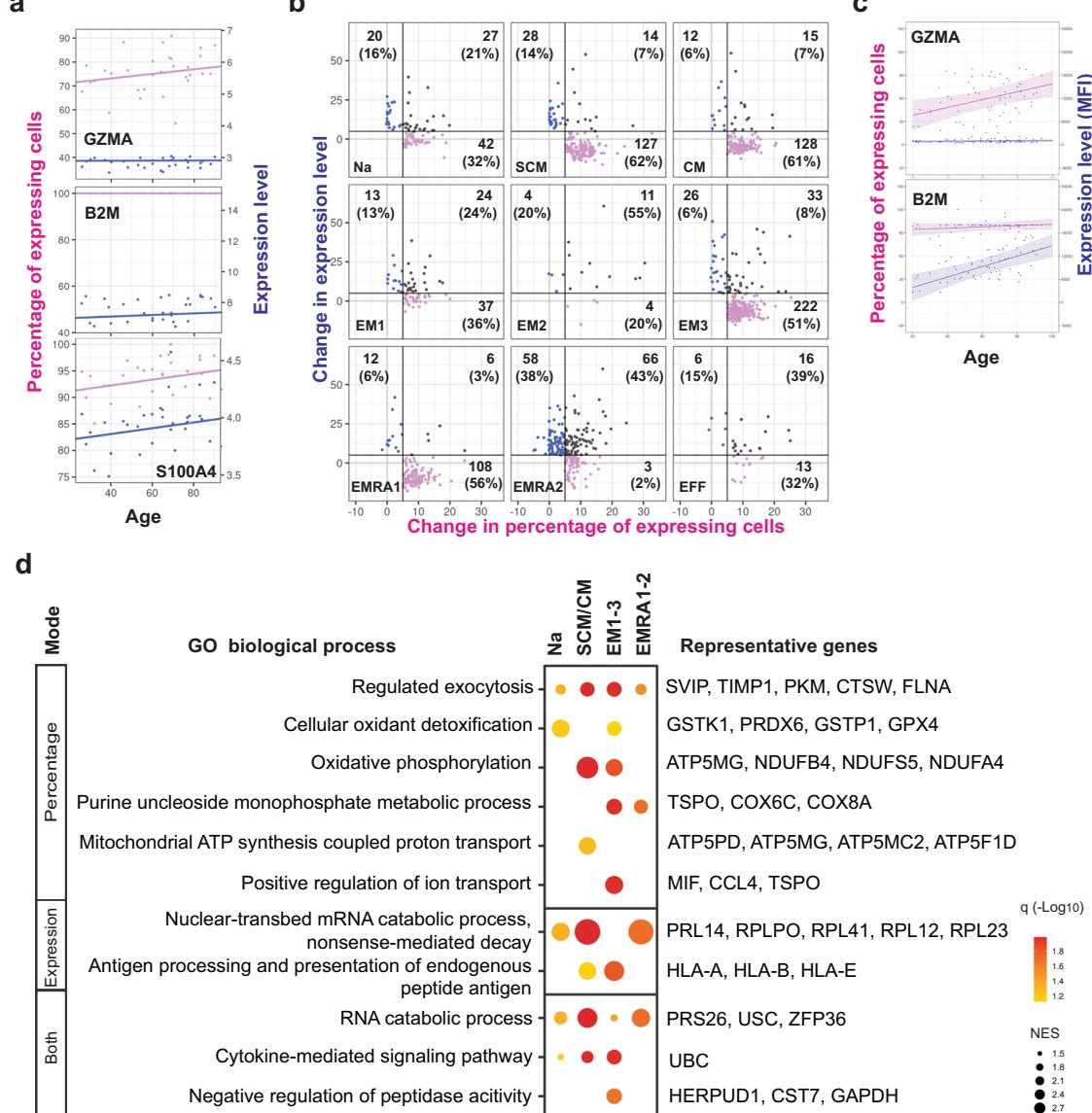

**Fig. 3 | Modes of age-associated increased gene expression in CD8⁺ T cell subpopulations. a** Three distinct modes of gene expression increase with aging in CD8⁺ T cells. A representative gene is shown for each mode. From top to bottom, GZMA increases in the number of expressing cells (represented by percentage in a donor); B2M increases by expression level in cells (represented by percentage increase in UMI counts); and S100A4 increases by number of expressing cells and by expression level. Trend lines were from the lineage regression model for which the t distribution with n-2 degrees of freedom was used to evaluate if a slope was significant for a particular variable (2 sided t-test) used here and in **c**.
**b** Classification of significant age-related increases in gene expression into three modes in each CD8⁺ T cell subpopulations (Na, naive adult; SCM, stem cell memory; CM central memory, EM1, 2, 3, effector memory subpopulations 1, 2, 3; EMRA1, 2

terminally differentiated effector memory cell subpopulation 1, 2, and EFF, effector cells) based on magnitude of change in percentage of detectable cells or average expression change in positive cells. The number and percentage of genes belonged to three modes were shown. Full gene list is in Supplementary Data 3. **c** Age-related mode of change at the protein level by flow cytometry. **d** Shared functional gene sets enriched in genes increasing with age in each of the three modes by gene set enrichment analysis (GSEA). The average significance defined by adjusted *p*-value (q) of each GSEA functional gene set analysis is represented in the heatmap in each cluster of CD8⁺ T cells by color scale. NES, normalized enrichment score. The scale of the magnitude of significance is 0 to 10⁻⁵. GO, gene ontology.

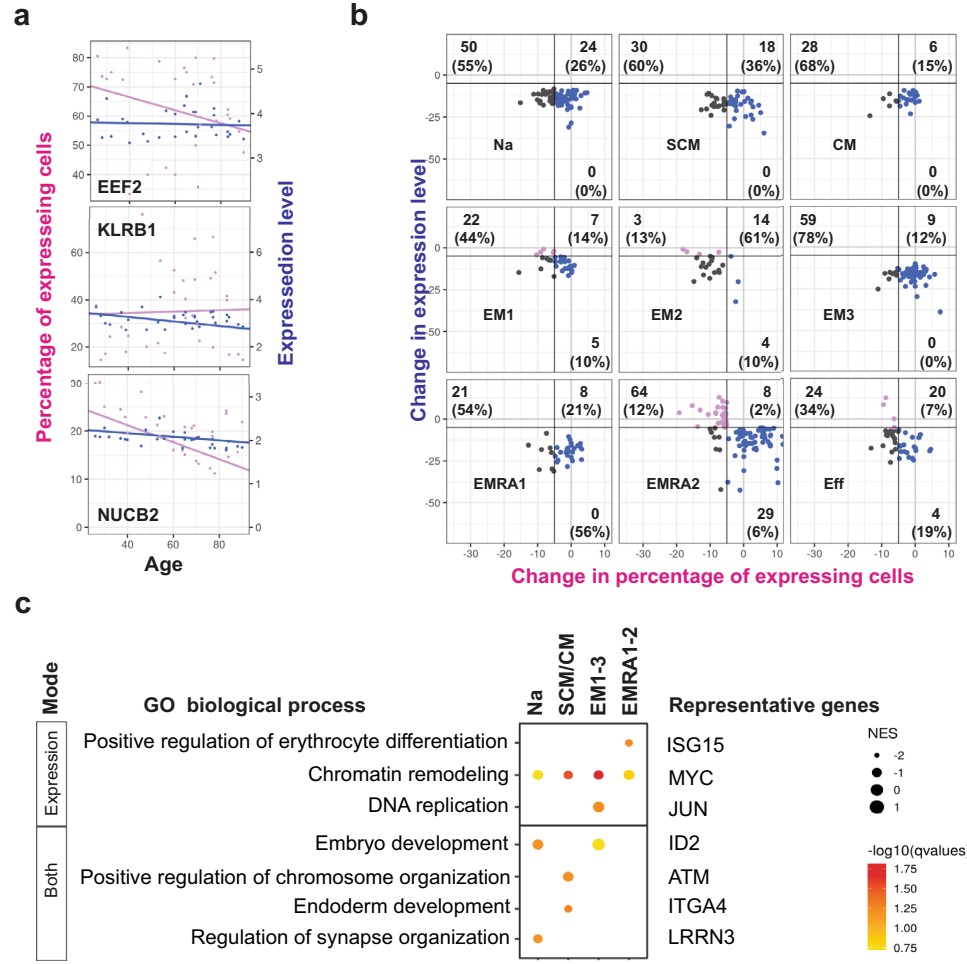

**Fig. 4 | Modes of age-associated reduced gene expression in CD8+ T cell subpopulations. a** Scatter plots showing examples of gene expression that decrease across aging in each of three modes (EEF2 – percentage, KLRB1 – expression, NUCB2 – both). Trend lines were from the lineage regression model for which the t distribution with n-2 degrees of freedom was used to evaluate if a slope was significant for a particular variable (2 sided t-test). **b** Scatter plots categorizing genes decreasing across age into three modes of change in nine clusters identified by scRNA-seq (Full gene list is in Supplementary Data 4). **c.** Bubble plot showing functional gene sets enriched for each of three modes of change across age. Representative genes for each gene set shared across clusters shown. NES, normalized enrichment score. The scale of the magnitude of significance is 0 to 10⁻⁵. GO, gene ontology.

immunotherapy, as both are involved the proliferation of CD8+ T cells[37,38]. Kazer et al. showed that HIV-1 infection induces high proliferation of CD8+ T cells[39] and our MEEN model found one year of HIV-1 infection-associated CD8+ T cell proliferation caused an average increase of 3.9 years in the predicted age of naïve CD8+ T cells (Fig. 7a, Supplementary Data 7a, b). Sheih et al. reported that infused CAR-CD8+ T cells in vivo undergo early rapid expansion and later stable maintenance[40]. We found a parallel rapid aging of CAR-CD8+ T cells at the expansion phase (28.6 ± 5.8 years) followed by a slowed pace of aging at the maintenance stage (5.2 ± 4.2 years) (Fig. 7b, Supplementary Data 7a, b). Collectively, these findings showed that the core age-related transcriptome changes in CD8+ T cells were influenced by cell divisions, as evidenced by the differentiation status and accumulation of mutations recognized by our MEEN model. Furthermore, predicted CD8+ T cell age was able to quantify aging as the in vivo proliferation of CD8+ T cells under different clinical conditions.

## Discussion

In this study, heterogeneity of human CD8+ T cells and their changes with age were analyzed by two single-cell methods (scRNAseq and high-dimensional flow cytometry) using both cross-sectional and longitudinal data. Our analyses identified 11 CD8+ T cell subpopulations and indicated previously unknown subpopulations within T_EM

and T_EMRA and their distinct changes with age. By integrating longitudinal and cross-sectional data using mixed-effect modelling to remove confounding person-to-person variation, we observed that most subpopulations of CD8+ T cells underwent substantial changes with age. Intriguingly, proportions of some subpopulations of memory (T_CM and T_EM2) and effector (T_EFF) cells were stable over 7 decades of life. How these age-associated changes for CD8+ T cells are modulated and what are their physiological significance remains to be determined.

Our analysis of the intrinsic changes in the transcriptome with age in subpopulations of CD8+ T cells showed some common changes, with more genes increasing than decreasing with age. GSEA analysis showed a range of altered functions associated with these changed gene expressions in T cell activation, antigen processing and presentation, and cytotoxicity. These data provide a basis for further studies to investigate the precise functional differences of these CD8+ T cell subpopulations and changes with age. Previous studies analyzing age-related gene expression changes in immune cells primarily reported average differences in population of cells between young and older adults. They lacked details about heterogeneity at the individual cell level. Based on these earlier studies, distinguishing the mechanisms by which these age changes occur at the individual cell level is not possible.

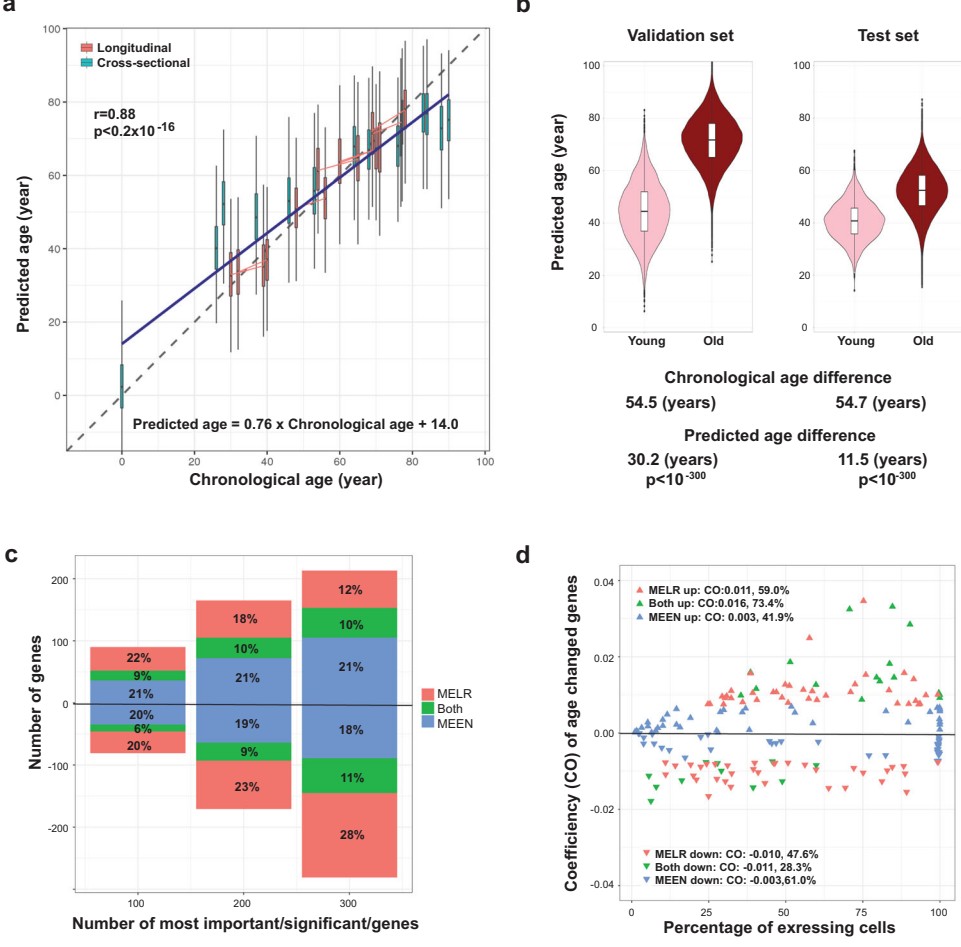

**Fig. 5 | Estimating the age of individual CD8+ T cells based on their transcriptomic signatures. a** Cross-validation age predictions for CD8+ T cells from each participant based on scRNA-seq data using a fitted mixed-effect elastic net (MEEN) model. Distribution of predicted cell ages for each donor is shown by violin plot (blue, cross-sectional donors; red, longitudinal donors). Donors' chronological age is on the x-axis. Trend lines were from the lineage regression model for which the t distribution with n-2 degrees of freedom was used to evaluate if a slope was significant for a particular variable (2 sided t-test). The data are in Supplementary Data 5a. **b** Left, cross validation of predictions for donors older than 70 years (Old) and younger than 30 years (Young). Right, test predictions for an independent dataset from healthy donors older than 70 years (Old) and younger than 30 years (Young) using public scRNA-seq datasets for CD8+ T cells. The box-whisker plots in

**a** and **b**: the center line is the median; the box is from the 25th to the 75th percentile. The upper or lower whisker extends from the hinge to the 1.5 x IQR (distance between the first and third quartiles) from the hinge for up and low, respectively. *P* values were calculated by 2-sided T-test. The data are in Supplementary Data 5b. **c** Overlap of age-associated genes in CD8+ T cells between MELR and MEEN models. The percentages of overlaps were incremented by the top 100, 200, and 300 genes identified by MELR based on FDR and coefficient of age. Percentages are grouped by if the genes were up or downregulated across aging as identified by the MELR method. The data are in Supplementary Data 5c. **d** Scatter plot of the coefficient of age (CO) of age-associated genes identified by MELR and the percentages of cells expressing each gene. The average change (CO) and percentage of each six groups are presented.

Here our analysis implicated age-related gene expression changes that could be regulated by three distinct mechanisms that could be associated with functional changes with aging. A change in the number of expressing cells suggests that aging may work as an on-off switch, while a change in the expression level within individual cells suggests that some genes are more finely tuned over time. Limited protein expression analysis by flow cytometry confirmed these modes of age-associated changes in CD8+ T cells. Furthermore, different modes of age-associated changes in gene expression appeared to be linked to distinct functions: increasing gene expression levels was linked to antigen presentation, whereas increasing numbers of cells expressing a gene was associated with cellular functions such as regulated exocytosis and oxidative phosphorylation. Investigating the regulators that cause these precise age-associated gene expression changes and their consequential alteration of functions in individual CD8+ T cells will lead to a better understanding of how age mechanistically alters the function of individual CD8+ T cells, paving the way for developing future novel therapeutic targets.

ML methods have recently been used to track biological aging[33,41]. These approaches have used both supervised and unsupervised models, leveraging multiple collected datasets such as plasma protein content, bulk RNAseq, and flow cytometry. The advantages of these methods include the ability to relate age predictions to clinical comorbidity and mortality. However, these methods have focused on understanding the overall age of the immune system. Here, we developed a MEEN model that predicts the age of single CD8+ T cells using the rich transcriptomic data of scRNAseq. We achieve a high correlation with chronological age assigned to cells by donor age in both our study and an independent dataset using both longitudinal and cross-sectional data. This transcriptome changes in biological aging at the cell level are strongly related to cell division because: 1) predicted cell age correlated with differentiation ($T_{Na}$ cells were consistently predicted to be younger than the more differentiated $T_{EM}$ and $T_{EMRA}$ subpopulations); and 2) mutation burden positively correlated with predicted cell age within CD8+ T cells. It is of great interest to further understand whether the mutation rates differ between the

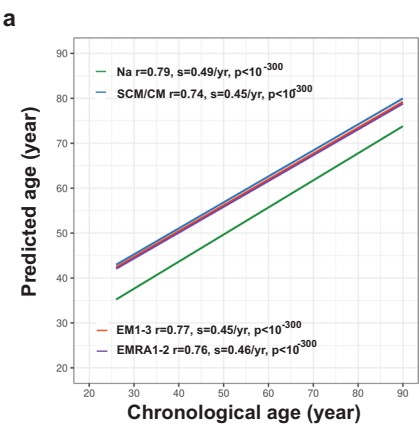

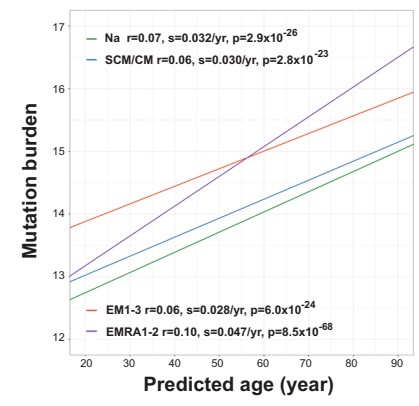

**Fig. 6 | Age-related transcriptomic changes associated with cell differentiation and divisions. a** Linear regression lines representing the cross validation predicted age for each of four major CD8 + T cell subsets by chronological age of donor. **b** Linear regression representing the relationship between mutation burden and cross validation predicted age of single cells in four major subsets of CD8+ T cells. Trend lines were from the MELR model for which the t distribution with n-2 degrees of freedom was used to evaluate if a slope was significant for a particular variable (2 sided T-test). The line data are in Supplementary Data 6a, b.

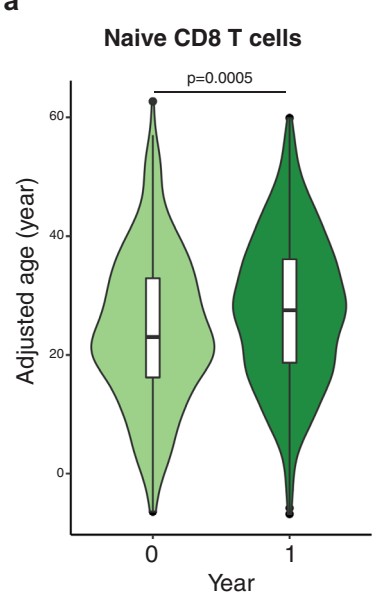

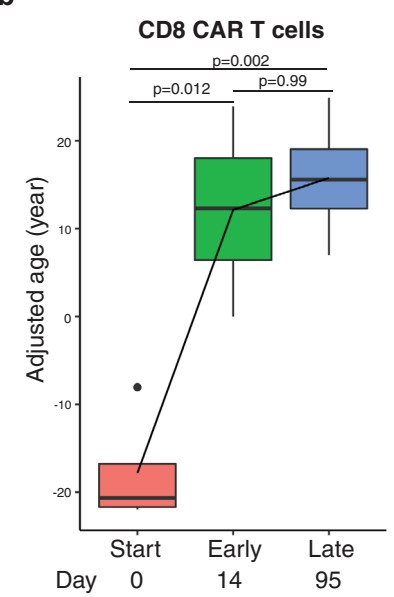

**Fig. 7 | Aging of CD8+ T cells under clinical conditions quantified by the ML algorithm. a** Predicted age increase in naïve CD8+ T cells during a one-year course after diagnosis of HIV-1 infection (blood sample collection from HIV-1 infected patients and used for scRNAseq analysis[39]). Predicted CD8+ T cell age at diagnosis and one year later (chronological increase in age was subtracted) based on Kazer et al.[39]. **b** Predicted age increase in CAR T cells during the course of treatment (CAR T cells collection at early and very late time points during the course of treatment and used for scRNAseq analysis[44]). Predicted CD8+ CAR T cell age prior to infusion and in post-infusion early phase (mean 14, range 12-21 days) and late phase (mean 95, range 83-112 days) based on Sheih et al.[40], $N = 4$. The box-whisker plots in **a** and **b**, the center line is the median, the box is from the 25th to the 75th percentile. The upper or lower whisker extends from the hinge to the 10% and 90% from the hinge for up and low, respectively. P values were calculated by two-sided T-test. The data are in Supplementary Data 7a, b.

processes of T cell development and differentiation, and between young and aged donors. Finally, the predicted cell ages of CD8+ T cells derived from clinical data under in vivo settings showed a good correlation with progressive HIV-1 infection and the degree of CAR T cell expansion during therapy as how rapid aging of CD8+ T cells can inform the clinical outcomes. These results demonstrate the ability of our ML model to detect biological changes in vivo. Most important, the power to predict the age of CD8+ T cells opens further possibilities for using transcriptomic cell age as a surrogate marker for aging of immune cells at the single cell level and the possibility of predicting clinical outcomes for which the function of CD8+ T cells is a determining factor.

## Methods
### Human participants

All studies on humans were approved by the National Institutes of Health Institutional Review Board and the Johns Hopkins Bayview Institutional Review Board. Written informed consent was obtained from all participants and complied with all relevant ethical regulations. Three cohorts were enrolled in this study: 1) A cross-sectional cohort of 14 healthy adults and 2 newborns with matched sexes in specific age ranges, and an overall age range from newborn to 90 years old. 2) A longitudinal cohort of 8 individuals with matched sexes. From this cohort, each participant made two or three visits to donate blood and the age range at first visit was 30-69 years old with an average of 9 years

between the first and second visit. Samples from the first and second cohorts were used for scRNAseq (Supplementary Table 1). 3) A cross-sectional cohort of 165 individuals ranging in age from 0 to 96 years (Supplementary Fig. 1a, Supplementary Data 1).

## Isolation of CD8⁺ T cells

Peripheral blood mononuclear cells (PBMCs) were isolated from human blood by Ficoll density gradient centrifugation. For adult donors in the cross-sectional cohort, CD8⁺ cells were enriched from PBMCs using an EasySep Human CD8 Positive Selection Kit II (STEM-CELL Technologies) according to the manufacturer's instructions. Cells were resuspended in phosphate-buffered saline (PBS) containing 0.04% bovine serum albumin (BSA). Purity was over 95% for all samples. PBMCs from cord blood were cryopreserved and thawed on the day of use in warm media for 1 h, washed once with BSM (HBSS containing 0.2% BSA, 1X HEPES, 1X penicillin-streptomycin-glutamine), and stained with fluorescent antibodies against CD3, CD4, and CD8. CD8⁺ cells were sorted as CD8⁺/CD4⁻ cells into PBS containing 0.04% BSA. For longitudinal cohort samples, cryopreserved PBMCs were thawed on the day of use in warm media for 1 h, washed once with BSM, and stained with antibody cocktails containing fluorescent antibodies against CD8 and CD4, and DNA barcoded antibodies against CD28 and CD45RA (Supplementary Fig 1c). CD8⁺ cells were sorted as CD8⁺/CD4⁻ cells into PBS containing 0.04% BSA. Purities of CD8⁺ T cells ranged from 97% to 98%. For scRNA-seq library construction, cells were diluted to ~1 ×10⁶/mL in 100 μL PBS containing 0.04% BSA and processed within hours after they were obtained.

## Single-cell RNA-seq library construction and sequencing

ScRNAseq libraries were constructed using Single Cell 3' library and gel bead kits (10X Genomics) following the manufacturer's protocol. Freshly enriched/sorted CD8⁺ T cells were kept on ice before processing. Viable cells were counted and ~8700 cells were used for gel bead-in-emulsion (GEM) generation (targeted recovery ~5000 cells). After template-switching reverse transcription within GEMs, cDNA was amplified in bulk. Amplification was followed by fragmentation, end repair, A-tailing, adaptor ligation, sample index PCR-generated P5- and P7- capping, and cell- and sample-indexing to complete the single-cell 3' gene expression libraries. For antibody (anti-CD45RA and CD28)-derived tag (ADT) libraries, cDNA was amplified for two rounds using KAPA HiFi PCR master mix (Roche) and indicated primers (Supplementary Table 3). PCR product was purified using 1.6X AmpureXP. Libraries were quantified using both a bioanalyzer machine (Agilent) and a Kapa library quantification kit (Roche) before sequencing. Sequencing was performed on an Illumina HiSeq 3000/4000 flow cell. Read 1, Read 2, and the sample index were sequenced to 28, 91, and 8 base pairs, respectively. Sample sequencing depth is summarized in Supplementary Table 1.

## scRNAseq data analysis

**Raw data processing.** Software (10x Genomics Cell Ranger 4.0) was used to analyze raw base calls[42]. Cell Ranger MKFASTQ was used to demultiplex raw base calls by sample index. Cell Ranger COUNT was used to align sequencing reads to the reference genome (10X reference 2020-A, GRCh38/Ensembl 98) and obtain unique molecular identifier (UMI) counts for each individual sample after merging multiple sequencing runs for the same library. Last, Cell Ranger AGGR was used to pool individual samples, and no-depth normalization was performed at this step. Analysis of these data was conducted using R 3.6.2. The output from Cell Ranger AGGR was analyzed by the R package Seurat V3.0.0[43] as a gene by cell barcode matrix. To remove low-quality cells and doublet cells, barcodes with UMI counts above 2500 and below 500 were removed. To remove broken and lysed cells, barcodes with mitochondrial DNA content greater than 20% were removed. Cells were clustered and projected onto UMAP components

1 and 2. Cell clusters with positive CD14, CD4, and CD19 or with no expression of CD3E, CD8A, and CD8B were filtered out.

**Integration of scRNAseq datasets.** Data were log₂-normalized separately for cross-sectional and longitudinal sample datasets. The results were then integrated − or batch-corrected − applying the standard integration pipeline implemented in Seurat[44] with the cross-sectional study as the "reference" dataset due to its larger size. Since cord blood was not present in the longitudinal study, cord blood samples were removed from the integration. Cord blood samples were replaced in the dataset after integration, as integration did not change expression values in the reference dataset. This integrated dataset was used for all further analysis.

**Clustering.** For clustering purposes, the integrated dataset was further integrated between cells from males and females to remove sex differences. The 2000 most variable genes were chosen using the FindVariableFeatures function in Seurat using the "vst" method. Clustering was performed in Seurat with 15 principal components at 0.9 resolution using the Louvain Algorithm. Cord blood samples clustered separately from adult cells in the dataset, creating two integrated clusters. These two clusters were merged and labelled "cord blood naïve" cells (N_C) and another six smaller clusters were also merged using a method similar to 'split-and-merge' methods[45]. Briefly, smaller clusters were merged based on gene expression similarity determined by hierarchical clustering and identified marker genes. A t-distributed stochastic neighbor-embedding (t-SNE) plot was used to represent cells in the dataset in two dimensions and identified cell clusters were labelled by color.

**Differential expression.** Differential expression analysis was performed in each cluster using Seurat using unintegrated data, while retaining the cluster identity. Genes with at least at an average 0.1 log2 fold-change (log2FC) in expression were considered differentially expressed at a threshold of 5% false discovery rate (FDR) generated from p-values using the Student's t-test. Heatmaps for expression of identified differentially expressed genes in each cluster were generated in the JMP statistical program (SAS) using a gene-by-cell-barcode matrix including selected genes from differential expression analysis, with cells ordered by cluster and age. Cell identities were assigned based on a combination of canonical marker genes and unpublished microarray studies performed on isolated CD8⁺ T cell subsets.

**Isoform identification and protein quantitation from ADT libraries.** CD45RA expression was confirmed by UMI counts in cells using ADT libraries for the CD45RA isoform of the CD45 gene that were generated for longitudinal samples. The mean CD45RA UMI count was calculated for each cluster. CD28 expression was quantitated using the same method.

## Multicolor flow cytometry analysis and subpopulation correlation analysis

The 15-color panel of antibodies (Supplementary Table 4) was against CD3, CD8, CD45RA, CD62L, CD95, CD27, CD28, CD69, CD2, CD127, GZMB, KLRG1, perforin, EOMES, and PD-1. Fixable Viability Stain 780 (BD Biosciences) was used to gate viable cells and 2 ×10⁶ PBMCs were stained with the surface antibody cocktail in 100 μl Brilliant Stain Buffer (BD Biosciences) for 30 min at 4 °C in the dark. After washing once with a FACS buffer (HBSS containing 0.2% BSA and 0.05% NaN₃), cells were fixed and permeabilized in Fixation/Permeabilization solution (BD Biosciences) for 20 min at 4 °C in the dark. Following fixation/permeabilization, cells were washed twice with 1x Perm/Wash buffer (BD Biosciences). Intracellular staining was in 1x Perm/Wash buffer for 30 min at 4 °C in the dark. Samples were washed twice with 1x Perm/Wash buffer and resuspended in 1% paraformaldehyde in PBS. Data

were acquired on a BD FACSymphony flow cytometer (BD Biosciences) and results were analyzed with FlowJo (10.3), which includes a plugin for FlowSOM software[34]. The Spearman's correlation coefficient for 13 marker genes (SELL, FAS, CD28, CD27, PDCD1, IL7R, KLRG1, CD2, EOMES, GZMB, PRF1, CD3E, and CD69) was calculated between clusters identified by FlowSOM and clusters identified by Seurat were computed to generate a correlation matrix[34]. This matrix was sorted by hierarchical clustering to match the most similar clusters between datasets.

## Gene set enrichment analysis

Gene set enrichment analysis (GSEA) was performed using the R packages clusterProfiler and ReactomePA on genes ordered by indicated measures in defined analyses[43]. GSEA annotation sets with normalized enrichment scores (NES) above 1.5 and q-value (FDR) below 10% were considered significant.

## Cluster distribution across age

The percentage of each cell cluster was determined for each donor sample. Linear mixed-effect regression (described below) was performed to fit the relationship between cell percentage of each donor and their chronological age while controlling for sex by including it as a covariate in the model.

## Gene expression changes across age

Linear mixed-effect modeling was used to identify upregulated and downregulated genes for each cluster across aging using the R package lme4[46]. This modeling method performs linear regression with a fixed-effect term and a random-effects term. The fixed-effect term measures how an independent variable affects an outcome. The random-effect term controls for differences inherent between multiple groups of data – in this case grouping cells by donor to isolate the longitudinal age-related changes for each donor. In our model, the age of each donor was used as a numerical independent variable and the log2-normalized UMI counts for each gene in each cell as dependent variables (outcomes), with the following Eq. 1:

$$UMI\_Expression \sim Age + Sex + (1|group) \qquad (1)$$

In this formula, Age was treated as a fixed-effect term and Sex was included as a covariate to control for sex differences. The (1|group) term represents a random-effect term, with cells grouped by the donor in the longitudinal study. Cells from all samples in the cross-sectional study were grouped into one group to remove longitudinal random effects. Genes that increased or decreased with a fixed effect magnitude larger than 0.0019 (10% per cell across the age range of the cohort) and FDR less than 5% were considered significant.

**Analysis of modes of gene expression changes**. Significantly upregulated or downregulated genes across age for each subpopulation were analyzed for the mode by which gene expression changed. The set of all genes with significant expression changes was classified by linear mixed-effect regression analysis using two different measures: the percentage of cells for each donor that expressed a particular gene, and the average expression of the gene in cells that measured positive for that gene. The fixed effect terms were age and sex, while the random effects term was one grouped by the donor as above. Positive cells were identified using the raw UMI counts matrix. Gene changes were labeled as either percent change, expression change, or both, if they exceeded a threshold value of 5% across the age range of the donors in either one measure or both. Genes that did not change by a magnitude of at least 5% in either measure were excluded from the analysis.

## Cellular age estimation

Models to estimate – or potentially predict - cellular age were constructed using linear and nonlinear approaches. For all models, a mixed-effect machine learning strategy was used based on Hajjem, Bellavance, and Larocque[47]. In these models, cells from cross-sectional donors were treated as one group and cells from longitudinal donors were grouped by the donor as above. This grouping allowed the models to integrate both cross-sectional and longitudinal patterns in gene expression when estimating cellular age.

To generate a linear model to predict cell age, the linear mixed-effect approach was used with elastic net-penalized multivariate regression in the R package glmnet[48] using all log2-normalized gene expression values as predictors of each cell's age. Elastic net-penalized regression was chosen based on its performance with sparse data in which the number of features (genes) is much larger than the number of observations (cells)[49]. Cross-validation was performed across 10 folds and for each fold, a random 90% of cells were used as a training set and 10% were used as a testing set across 100 lambda values; this step determined how many variables were removed from each model. The lambda value yielding the lowest mean square error was chosen for the linear model penalty value.

The nonlinear mixed-effects approach used the R package ranger to generate a random forest using the same input covariates as the linear model[50]. The model was generated using 500 iterations with a bootstrapping method to select training and cross-validation sets for each individual tree. Briefly, the bootstrapping method involved selecting the same n number of cells as in the dataset with the replacement for each iteration. Individual trees were trimmed to have a minimum of 100 observations per node split in each tree. Sex was included in all models generated as a one-hot encoded covariate, i.e., as 1 s (male) and 0 s (female) to control for sex differences.

For testing our mixed-effect predictive models on unseen data, two datasets were downloaded: CD8[+] T cell and PBMC scRNAseq data from 10X genomics and data from the European Genome-Phenome Archive (accession EGAD00001006000)[51]. Data in the 10X genomics datasets was parsed, and using UMAP projections, clusters were identified that expressed CD8B, CD8A, and CD3E. Cells were filtered from this dataset using the strategy above. Data were log2-normalized, and no further changes were made before generating predictions. The scRNAseq datasets from HIV-1 infection[39] and CAR T cells[40] were downloaded from the Gene Expression Omnibus (GEO). Predictions were generated by extracting CD8A + and CD8B + cell clusters using the provided metadata in the deposition on GEO. Cell identity was verified by the expression of CD8A, CD8B, and CD3E. Cells were filtered from this dataset using the strategy above. Data were log2-normalized. Predictions were made directly on these data using the models generated above with no further changes to the dataset.

To estimate the importance, or relative contribution, of genes to the predictions made by our models, we used the R package DALEX to generate variable importance values for each gene. Briefly, the DALEX package has a permutation-based algorithm that permutes the value of each variable and assesses the increase in error compared to the baseline prediction[52].

## Exome sequencing

Cryopreserved PBMCs were thawed, and genomic DNA was prepared from 1–2 million PBMCs using the DNeasy Blood & Tissue Kit (QIAGEN). More than 0.5 mg genomic DNA was used for whole exome sequencing at Novogen using an exome sequencing kit (Twist Biosciences). Briefly, the protocol enriched sequences in the defined CCDS region[50] and targets were reverse transcribed and sequenced. Sequencing was on an Illumina NovaSeq 6000 with paired-end 150 bp reads and effective sequencing depth above 50X.

Preprocessing of raw exome sequencing files was performed by aligning reads to the reference genome (GRCh38) using BWA mem.

Single nucleotide polymorphisms (SNPs) for each donor were generated using GATK haplotypecaller and hard filtered using the criteria in the recommended best practices. The reference genome was modified for each donor to include that donor's SNPs in the consensus reference. These donor-specific references were used in our variant-calling pipeline.

## Variant calling and UMI correction

Variant calling was performed on each cell using our custom pipeline developed to remove errors in the scRNAseq dataset using a UMI-based correction strategy[53]. This pipeline uses the GATK mutect2 program and is modified to use donor-specific references that include SNPs for each donor. This pipeline is available for use on GitHub (https://doi.org/10.5281/zenodo.4924437 https://github.com/Weng-lab-NIH/USCMD). Coordinate-sorted STAR-aligned reads from Cell Ranger AGGR output were separated into individual BAM files for each cell for each donor sample. Variant calling was performed using GATK Mutect2 to call somatic mutations, following the best practices for somatic variant calling[36] using the donor-specific references for the reference option. Single cells were considered tumor samples in the pipeline and exome sequencing reads for individual donors were used as normal samples in a BAM file. Variants for each single cell were filtered for those with tumor log-odds ratio (TLOD) > 5.3 (default for Mutect2) and depth in exome sequencing files >10X. Raw variants were filtered for SNPs, removing indels. UMI-based correction was performed by analyzing if SNPs called were represented by at least three supporting reads within a UMI barcode and if the percentage of supporting reads was more than 50% for a variant within a UMI barcode. Positions with more than 2 mutations that passed the filters were removed since we only expected up to two mutated alleles. To calculate each single cell's normalized mutation number, the number of mutations within each cell was log2 transformed. Then, a negative binomial model was generated, using the UMI number in a cell, coverage in a cell, and coverage in the associated donor's exome as covariates to control for these factors in the cell's mutation count. Adjusted mutation counts were then scaled from 0 to the maximum number of raw mutations detected in a cell in the dataset to restore the original range of values.

## Reporting summary

Further information on research design is available in the Nature Research Reporting Summary linked to this article.

# Data availability

The scRNAseq data are available at NCBI: GSE136184 (https://www.ncbi.nlm.nih.gov/geo/query/acc.cgi?acc=GSE136184), and EGAD00001006000 (https://ega-archive.org/datasets/EGAD00001006000). Source data are provided with this paper.

# Code availability

The script https://doi.org/10.5281/zenodo.4924437 files to perform all analysis to reproduce the data/results in the paper as well as recreate all figures are deposited at GitHub (https://zenodo.org/badge/latestdoi/364361401), https://doi.org/10.5281/zenodo.6473570, and the somatic mutation identification pipeline (USCMD) is deposited at https://zenodo.org/badge/latestdoi/390409912 and https://doi.org/10.5281/zenodo.5705233.

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

## Acknowledgements

We thank Richard Hodes and Jiang Peng for critical reading and comments on the manuscript, Kimberly Jones-Beatty, and Mia Feller for helping collecting cord blood samples, and Christopher Dunn, Cuong Nguyen and Tonya Wolf for assistance with flow analysis and cell sorting. This work used the computational resources of the NIH HPC Biowulf cluster and was supported by the Intramural Research Programs of the National Institutes of Health National Institute on Aging and National Heart, Lung, and Blood Institute.

## Author contributions

The concept and study design were conceived and developed by J.L., R.A. and N.P.W. The wet lab experiments were performed by J.L., J.Y.L., JinguoC, S.L., A.A., M.C., A.L., J.Z., and C.D. The computational analyses were conducted by R.A., T.N., JeffreyC, H.H., J.W., JosephC. Samples and logistic arrangements were assisted by L.Z., C.C., I.B., and L.F. Manuscript was written by J.L., R.A., L.F. and N.P.W. with approval from all authors.

## Funding

## Competing interests

The authors declare no competing interests.
