## [Peer Review File · Nature Communications]

Heterogeneity and transcriptome changes of human CD8+ T cells across nine decades of lifeREVIEWER COMMENTS

Reviewer #1 ageing in the immune system (Remarks to the Author):

This manuscript contains an impressive amount of data looking at changes in T cell (subsets) in people with different ages. The main conclusions are that different subsets of CD8+ T cells can be identified using either transcriptome or flow analysis and these subsets changes with age. Further, age-related gene expression can be due to changes in the relative amount of subsets, changes of gene expression within cells or a correlation of both. Using their single cell data set the authors used ML models and mutation analyses to predict cell age and found that naïve T cells were consistently younger than antigen-experienced cells.

My main concerns are:

- Although very well performed and using a different cohorts the fact that (1) different subsets of CD8+ T cells exist, that naïve T cells in the blood go down with age whereas cells that express effector molecules go up is not novel.
- It is generally expected that herpes viruses like EBV and CMV have huge impact on T cell subset distribution and (cytolytic) effector molecule expression. It's unclear if the authors have stratified for these latent infections and if e.g. the drop in naïve T cell levels are being observed in e.g. CMV-negative people.
- This is an observational study that doesn't allow strong conclusions on the implications of the findings for clinical immunology and the conclusions in the discussion support this notion: "The physiological significance of these changes for CD8+ T cell function with aging remains to be determined." and "Uncovering the regulators that cause these precise age-associated gene expression changes and their consequential alteration of functions in individual CD8+ T cells will lead to a better understanding of how age mechanistically alters the function of individual CD8+ T cells, paving the way for developing future novel therapeutic targets."

Reviewer #2 systems immunology and machine learning (Remarks to the Author):

The authors present a combined cross-sectional and longitudinal single-cell study of CD8+ T cell aging and associated changes in cellular functions. By utilizing single-cell RNAseq and flow cytometry, the authors identified subsets of the cells investigated, described their modes of transcriptomic changes, and hypothesized potential causal relations to underlying changes in the cells.

The analysis is well designed with a solid data basis. Large data sets were generated from multiple donors virtually covering the whole life span. The manuscript is sound and reads well. However I have few comments which may help to further improve it and I look forward to seeing this manuscript after revision.

Major comments:

- In the text (lines 180-182), overall percentages for cells changing in the three modes or not are given. I propose to add an 'overall' plot to Figure 3b showing all cells and the numbers given in the text.
- Differentiation state and mutational burden involvement is important. In this context, please discuss the difference between chronological age and time since the cells emerged from their progenitor cells.
- What are your next steps in this topic, how can these results be exploited in the future?
- Please state why are CD8+ cells defined as CD3+CD19-CD8+CD4- (line 319), but later as CD8+CD14- (line 323).
- How were small clusters 'merged based on expression similarity (line 375)? Please indicate the algorithm or tool used here.

Minor comments:

- Figure 1a is a bit unclear and should be streamlined to better highlight the three cohorts and the two single cell approaches/data sets.
- Several established marker genes were used to assign subpopulations. I suggest to collect them in a table for lucidity and describe how they were selected (by hand from cluster differential gene list?)
- Listing of the subpopulations is slightly hard to follow. They could be formatted using bullet points, also the table mentioned above would help to easily get an overview about the subpopulations.
- Figure 3c: please use the same gene labels as in panel a.
- Line 282: reference format is broken
- Line 321: remove the '.' after 'hour'
- The list of PCS primer (lines 337-343) perturb readability of the corresponding section and should be moved to a table in supplementary material.
- Line 382: what does JMP and SAS stand for?
- Line 405: reference format is broken
- Figure 7a and c are not really informative and can be removed in my opinion.

Reviewer #3 immune ageing (Remarks to the Author):

The paper by Lu et al highlights the heterogeneity in the CD8 T cell compartment. While this is not new I did like the comparison between the change in percentage of gene expressing cells with the change in expression of genes, but they don't really go into the meaning of this. Intuitively it seems like changes in percentage represent age dependent processes that are switched on or off whilst expression changes are merely ramped up or down. They also develop a machine learning model capable of predicting the age of individual cells based on transcriptomic data, again this is not new and while they do reference IMM-AGE and iAGE the authors don't say how their model differs. The authors need to provide a full github script for analysis reproducibility rather than just on how to make the figures.

More specifically:

1. Page 7 (Fig 1f): Some subsets (Em1, EM3, and EFF) have low correlation between proteins and gene transcripts, is there a role for post-transcriptional regulation?
2. Page 7 (Fig 2a, b): Whilst the age-associated changes in CD8 subsets do look comparable between scRNAseq and flow cytometry, they need to point out in the manuscript that some of the correlations that were significant in the flow cytometry data were not in the scRNAseq data.
3. Page 10 (Fig 5a, b/Methods page 21): How was the train/test data split for the training and validation of MEEN and MERF models? If the test set is comprised of cells belonging to donors represented in the train set, then this runs the risk of overfitting. Specify how overfitting was mitigated.
4. Page 11: What are the relative contributions of genes to each of the models predictions? Could you use methods from explainable AI such as SHAP values to explain the relative contributions of the features to the models predictions?
5. Page 17: Was the differential expression analysis conducted on the integrated or the unintegrated scRNAseq data? Please specify this in the methods. Integration causes dependencies between data points and violates the assumptions of downstream differential expression analysis. The Seurat pipeline specifies that differential expression testing should be conducted on unintegrated data.
6. Page 20: In the analysis of modes gene expression changes specify the fixed and random effects used in each of the mixed effect regression analyses.

AUTHOR'S RESPONSE TO REVIEWERS

Reviewer #1

This manuscript contains an impressive amount of data looking at changes in T cell (subsets) in people with different ages. The main conclusions are that different subsets of CD8+ T cells can be identified using either transcriptome or flow analysis and these subsets changes with age. Further, age-related gene expression can be due to changes in the relative amount of subsets, changes of gene expression within cells or a correlation of both. Using their single cell data set the authors used ML models and mutation analyses to predict cell age and found that naïve T cells were consistently younger than antigen-experienced cells.

Response: We thanks Reviewer 2 for the supportive comments and constructive suggestions.

My main concerns are:

1. Although very well performed and using a different cohorts the fact that (1) different subsets of CD8+ T cells exist, that naïve T cells in the blood go down with age whereas cells that express effector molecules go up is not novel.

Response: We agree with this assessment. Here we aimed to highlight novel memory subpopulations that are present in blood and the different patterns of transcriptional aging changes in both naïve and memory populations.

2. It is generally expected that herpes viruses like EBV and CMV have huge impact on T cell subset distribution and (cytolytic) effector molecule expression. It's unclear if the authors have stratified for these latent infections and if e.g. the drop in naïve T cell levels are being observed in e.g. CMV-negative people.

Response: We acknowledge the highly heterogenous circulating CD8 T cells are products of their history and environmental experiences. The subjects in this study were healthy with no chronic illnesses or cancer, but we do not have their infection history for EBV, CMV, or other infectious agents. Therefore, we were not able to stratify by this data.

3. This is an observational study that doesn't allow strong conclusions on the implications of the findings for clinical immunology and the conclusions in the discussion support this notion: "The physiological significance of these changes for CD8+ T cell function with aging remains to be determined." and "Uncovering the regulators that cause these precise age-associated gene expression changes and their consequential alteration of functions in individual CD8+ T cells will lead to a better understanding of how age mechanistically alters the function of individual CD8+ T cells, paving the way for developing future novel therapeutic targets."

Response: We agree and believe our findings will serve as a foundation for further analysis to reveal clinical consequences of these observed changes. However, applications of our ML model of aging include two clinical cases (HIV infection and CAR-T cell therapy) with promising results, suggesting this ML model could be a useful tool for clinical settings where the health of CD8 T cells is relevant for cell-based therapy.

Reviewer #2

The authors present a combined cross-sectional and longitudinal single-cell study of CD8+ T cell aging and associated changes in cellular functions. By utilizing single-cell RNAseq and flow cytometry, the authors identified subsets of the cells investigated, described their modes of transcriptomic changes, and hypothesized potential causal relations to underlying changes in the cells.

The analysis is well designed with a solid data basis. Large data sets were generated from multiple donors virtually covering the whole life span. The manuscript is sound and reads well. However I have few comments which may help to further improve it and I look forward to seeing this manuscript after revision.

Response: *We thanks Reviewer 2 for the supportive comments and constructive suggestions.*

Major comments:

1a. In the text (lines 180-182), overall percentages for cells changing in the three modes or not are given. I propose to add an 'overall' plot to Figure 3b showing all cells and the numbers given in the text.

Response: *We have added suggested percentages in Fig 3b and Fig 4b now.*

1b. Differentiation state and mutational burden involvement is important. In this context, please discuss the difference between chronological age and time since the cells emerged from their progenitor cells.

Response: *This is an important issue that has implications in aging. Although our mutation data in CD8 T cells can't be used to distinguish between chronological age and time when the cells emerged from their progenitor cells, we have added a short note about this in the discussion (p14).*

2. What are your next steps in this topic, how can these results be exploited in the future?
- Please state why are CD8+ cells defined as CD3+CD19-CD8+CD4- (line 319), but later as CD8+CD14- (line 323).

Response: *1) Currently, we have two directions: one is to explore this age prediction in other types of lymphocytes such as CD4, B and NK cells (using ML transfer learning) and the other is to expand the age analysis in clinical settings to see if predicted age changes are associated with clinical outcomes. 2) This was our mistake and we have revised sort gating as both CD8 T cells were sorted by gating on CD8+CD4- T cells (p15) and the CD8 identity was further confirmed in scRNAseq cluster analysis as CD8+CD3+CD19-CD4-.*

3. How were small clusters 'merged based on expression similarity (line 375)? Please indicate the algorithm or tool used here.

Response: *Clusters were merged based on a strategy similar to 'split-and-merge'. Briefly,*

smaller clusters were merged based on gene expression similarity determined by hierarchical clustering (heatmap shown below left) and on the overlap of identified marker genes (table shown below right) (see Method on p18).

subset	original identity	original number	combined number	distinguishing genes
CB	6	8132	8644	HBA1, HBA2, SH2D1A, FXJD2, GIMAP7, LEF1, FCER1G
	19	492		
N	1	17637	22607	AIF1, TRABD2A
	10	4970		
SCM	5	10682	14258	SOCS3, GATA3, AP3M2
	12	3576		
CM	3	11985	11985	
EM1	2	15548	16477	TXNIP, CMC1
	17	929		
EM2	14	2044	2044	
EM3	8	7410	9189	NCR3, DUSP1, KLRC1, KLRB1, ZBTB16, CEBPD
	15	1680		
RA1	22	99	20427	
	0	20427		
RA2	7	7820	11996	CCL3, CCL4, KLRF1
	11	4176		
EFF	13	2427	2791	MTRNR2L8, HLA-A, HLA-B, MT-CO1, MT-CO2
	20	364		

Minor comments:

- Figure 1a is a bit unclear and should be streamlined to better highlight the three cohorts and the two single cell approaches/data sets.

Response: Revised as suggested.

- Several established marker genes were used to assign subpopulations. I suggest to collect them in a table for lucidity and describe how they were selected (by hand from cluster differential gene list?).

Response: We revised the description of gene selection and provided a bubble plot that shows percentage of expression within a cluster and expression level of genes that define the subsets in a revised Supplemental fig. 1d.

- Listing of the subpopulations is slightly hard to follow. They could be formatted using bullet points, also the table mentioned above would help to easily get an overview about the subpopulations.

Response: Revised Fig.1c and supplementary fig. 1d.

- Figure 3c: please use the same gene labels as in panel a.

Response: Done.

- Line 282: reference format is broken.

Response: Corrected.

- Line 321: remove the '.' after 'hour'

Response: Done.

- The list of PCS primer (lines 337-343) perturb readability of the corresponding section and should be move to a table in supplementary material.

Response: Done (Supplementary table 12).

- Line 382: what does JMP and SAS stand for?

Response: JMP is a statistical software made by the company SAS.

- Line 405: reference format is broken.

Response: Corrected.

- Figure 7a and c are not really informative and can be removed in my opinion.

Response: Removed as suggested.

Reviewer #3

The paper by Lu et al highlights the heterogeneity in the CD8 T cell compartment. While this is not new I did like the comparison between the change in percentage of gene expressing cells with the change in expression of genes, but they don't really go into the meaning of this. Intuitively it seems like changes in percentage represent age dependent processes that are switched on or off whilst expression changes are merely ramped up or down. They also develop a machine learning model capable of predicting the age of individual cells based on transcriptomic data, again this is not new and while they do reference IMM-AGE and iAGE the authors don't say how their model differs.

The authors need to provide a full github script for analysis reproducibility rather than just on how to make the figures.

Response: We thanks Reviewer 2 for the supportive comments and constructive suggestions. We have updated the description in our manuscript and GitHub page to reflect that the script files perform all analysis to reproduce the data/results in the paper as well as recreate all figures. We also updated the manuscript to describe how our model compares/contrasts with previous models related to aging in terms of the data used, model architecture, and prediction made (p14).

More specifically:

1. Page 7 (Fig 1f): Some subsets (Em1, EM3, and EFF) have low correlation between proteins and gene transcripts, is there a role for post-transcriptional regulation?

Response: Post-transcriptional regulation as well as posttranslational modifications are possible explanations as to why there is a discrepancy.

2. Page 7 (Fig 2a, b): Whilst the age-associated changes in CD8 subsets do look comparable between scRNAseq and flow cytometry, they need to point out in the manuscript that some of the correlations that were significant in the flow cytometry data were not in the scRNAseq data.

Response: *We have included this in the revised manuscript (p7-8) and the main reason for the difference in statistical significance is the number of subjects used in each method (24 for scRNAseq and 165 for flow cytometry).*

3. Page 10 (Fig 5a, b/Methods page 21): How was the train/test data split for the training and validation of MEEN and MERF models? If the test set is comprised of cells belonging to donors represented in the train set, then this runs the risk of overfitting. Specify how overfitting was mitigated.

Response: *The train/test split was performed on the cell level (not mutually excluding donors in train/test for model building). Although we did not split based on donor, we implemented the following two methods to prevent overfitting:*

1) Model Regularization: In both the MEEN and MERF models, we used high levels of regularization to produce more generalized models. For example, in the MEEN model, we use the elastic net method which uses L1 and L2 penalties to reduce coefficients for noisy and unreliable variables. In the MERF model, we used a high tree number (500) and a high terminal node split (100) to prevent overfit forests.

2) Final Model Selection Based on Test Prediction: On top of regularization, we implemented 2 models, a MERF and MEEN model. Using these two models, we tested each one on external data that was not sampled from our internal training/testing dataset. This dataset had a similar range of healthy donor ages, was generated from the same 10X 3' technology. Prediction on this test set allowed us to choose the model between the MEEN and MERF that was better able to distinguish old and young patients at the donor level. The MEEN generalized better, and therefore was overfit less on the train set. So, our final model for the rest of the paper was the MEEN model rather than MERF.

4. Page 11: What are the relative contributions of genes to each of the models predictions? Could you use methods from explainable AI such as SHAP values to explain the relative contributions of the features to the models predictions?

Response: *We have included variable importance values (Supplemental Table 9) generated from the DALEX R package, added two graphs to figure 5, added this analysis to the methods section (p22), and discussed in the results of the revised manuscript (p10).*

5. Page 17: Was the differential expression analysis conducted on the integrated or the unintegrated scRNAseq data? Please specify this in the methods. Integration causes dependencies between data points and violates the assumptions of downstream differential expression analysis. The Seurat pipeline specifies that differential expression testing should be conducted on unintegrated data.

Response: *We previously analyzed the differential expression on the integrated dataset, and now we have updated the analysis that was carried out on the raw data (retaining cluster*

identity). We updated the figure 1c and supplemental Fig. 1d and supplemental table 3 to reflect this change and added this to the methods (p18).

6. Page 20: In the analysis of modes gene expression changes specify the fixed and random effects used in each of the mixed effect regression analyses.

Response: *We added this to our methods (p21).*

REVIEWERS' COMMENTS

Reviewer #1 (Remarks to the Author):

It's a pity that data on herpesvirus status were not available for further analyses. This limits the appreciation of the importance of the data sets.

Reviewer #2 (Remarks to the Author):

The authors addressed all my comments adequately. I suggest to publish this manuscript.

Reviewer #3 (Remarks to the Author):

The authors have addressed all my questions and have included these where appropriate in the paper.